# Invasive Mechanical Ventilation and Death Was More Likely in Patients with Lower LDL Cholesterol Levels during COVID-19 Hospitalization: A Retrospective Propensity-Matched Cohort Study

Adhya Mehta [1], Amrin Kharawala [1], Sanjana Nagraj [1,*], Samuel J. Apple [1], Diego Barzallo [1], Majd Al Deen Alhuarrat [1], Cesar Joel Benites Moya [1], Sindhu Vikash [1], Panagiotis Zoumpourlis [1], Sophia Xesfingi [2], Dimitrios Varrias [1], Yunus Emre Demirhan [1], Leonidas Palaiodimos [1,3] and Dimitrios Karamanis [4,5]

[1]   Department of Medicine, Albert Einstein College of Medicine/Jacobi Medical Center, New York City Health + Hospitals, 1400 Pelham Parkway S, Bronx, NY 10461, USA; mehtaa10@nychhc.org (A.M.); kharawaa@nychhc.org (A.K.); leonidas.palaiodimos@nychhc.org (L.P.)
[2]   National Documentation Center, Zefirou 56, 17564 Paleo Faliro, Greece
[3]   CUNY School of Medicine, New York, NY 10031, USA
[4]   Department of Economics, University of Piraeus, 18534 Piraeus, Greece; dkaramanis@hotmail.com
[5]   Department of Health Informatics, Rutgers School of Health Professions, Newark, NJ 07107, USA
[*]   Correspondence: sanjana94nagraj@gmail.com; Tel.: +1-718-918-5645; Fax: +1-571-376-6710

**Abstract:** Hyperlipidemia has been associated with worse outcomes in patients with Coronavirus disease 2019 (COVID-19). However, lower LDL-C (low-density lipoprotein cholesterol) levels have been associated with increased COVID-19 severity and mortality. We conducted a retrospective observational study of patients with COVID-19 admitted to New York City Health and Hospitals from 1 March 2020 to 31 October 2020, comparing pre-COVID-19 LDL-C levels or LDL-C levels obtained during COVID-19 hospitalization, with the need for invasive mechanical ventilation and death. Propensity score matching was performed using logistic regression models, and standardized mean differences were calculated. A total of 3020 patients (median age 61 years; 36% women) were included. In the matched cohort, on multivariate logistic regression analysis, LDL was inversely associated with in-hospital death (OR: 0.99, 95% CI: 0.986–0.999, $p = 0.036$). As a categorical variable, LDL > 70 mg/dL was associated with 47% lower likelihood of invasive mechanical ventilation (OR: 0.53, 95% CI: 0.29–0.95, $p = 0.034$). No significant association between pre-COVID-19 LDL and death or invasive mechanical ventilation was found (OR: 1.00, 95% CI 0.99–1.01, $p = 0.833$). Low LDL-C level measured during COVID-19 was associated with a higher likelihood of invasive mechanical ventilation and in-hospital death. A similar association was not found between pre-COVID-19 LDL-C and these outcomes. LDL-C levels obtained during COVID-19 are likely not reflective of the baseline lipid profile.

**Keywords:** COVID-19; LDL-C; invasive mechanical ventilation

## 1. Introduction

Hyperlipidemia, like other cardiovascular risk factors, has been associated with worse outcomes in patients with Coronavirus disease 2019 (COVID-19) [1–8]. Therefore, it would be reasonable to hypothesize that high low-density lipoprotein cholesterol (LDL-C) level is a predictor of respiratory failure, the need for invasive mechanical ventilation, and death in patients with COVID-19. However, the existing literature does not necessarily support this hypothesis [9,10].

The presence of low levels of LDL-C in patients with COVID-19 has been associated with increased disease severity and higher mortality from COVID-19 [9,10]. In the meta-analysis by Mahat et al., pooled data from nineteen studies with COVID-19 patients found lower levels of total cholesterol (TC), high-density lipoprotein cholesterol (HDL-C), and LDL-C to be associated with worse outcomes [10]. Investigators have attributed this paradoxical

finding to severe inflammation, which results in altered lipid metabolism and lipid utilization by the virus for replication, membrane homeostasis, endocytosis, and exocytosis [9].

Importantly, the majority of these studies were conducted in populations outside the United States [10]. It is not clear whether a similar finding would be observed in our ethnically diverse and socioeconomically disadvantaged patient population, carrying a high cardiovascular disease risk burden and served by the New York City Health and Hospitals system. It is also unclear whether the lipid profile can be used as a reliable marker of COVID-19 severity and prognosis. Taking the above into account, we aimed to investigate and compare the associations between pre-COVID-19 and during COVID-19 LDL-C levels with the need for invasive mechanical ventilation and mortality in patients with COVID-19.

## 2. Material and Methods

### 2.1. Study Design, Study Setting, and Patient Population

We conducted a retrospective observational study on data collected from the eleven acute care hospitals of the New York City Health and Hospitals (NYC H and H) system, which is the largest public hospital system in the United States, and the largest safety net hospital system in the country [11]. NYC Health and Hospitals is the largest public healthcare system in the United States (US). It caters to nearly one million individuals annually, of which around 70% are people of color. The majority of patients treated belong to the low-income strata, with 32% of care seekers being uninsured, and 35% being Medicaid beneficiaries [11]. During the pandemic, NYC H and H expanded its capacity to incorporate more patients. By March 2021, H and H had treated more than 108,000 COVID-19 patients, and more than 54,000 hospitalized patients with COVID-19 had been discharged [12]. The duration of the study was from 1 March 2020 to 31 October 2021. Our study duration incorporates the earlier stages of the pandemic, the COVID-19 surge, the rise of new variants, and the period of initiation of vaccines against COVID-19 infection. New York was one of the worst affected states in the US. At one point, the number of infected far exceeded the capacity of the hospitals, and multiple field hospitals had to open.

Patients $\geq$ 18 years of age who presented to the emergency room and were admitted to any inpatient service, including the intensive care unit (ICU), with laboratory-confirmed COVID-19 and available LDL-C level, either during the index admission or within six months prior to index admission, were included. Laboratory-confirmed COVID-19 was defined as a SARS-CoV-2 positive result in real-time reverse transcriptase polymerase chain reaction (RT-PCR) analysis of nasopharyngeal or nasal swab samples. We excluded patients who met $\geq$ 1 of the following criteria: (i) patients < 18 years old; (ii) patients without laboratory-confirmed COVID-19; (iii) women who were pregnant at the time of the index hospitalization.

This study was approved by the Biomedical Research Alliance of New York Institutional Review Board, with a waiver of informed consent (IRB #20-12-103-373). Data were fully deidentified and anonymized before they were accessed, hence the IRB waived the requirement for informed consent.

### 2.2. Data Sources

Study data were obtained from electronic health records via appropriate International Statistical Classification of Diseases (ICD) codes (Epic systems, Verona, WI, USA). The initial dataset was reviewed by two independent investigators for accuracy. Three pairs of additional independent investigators reviewed individual charts when clarifications were needed. The extracted data included age, sex, race, ethnicity, body mass index (BMI), history of tobacco use, hypertension, lipid profile, diabetes, coronary artery disease (CAD), heart failure, stroke, peripheral artery disease, atrial fibrillation, chronic obstructive pulmonary disease (COPD), asthma, chronic kidney disease (CKD) and end-stage renal disease (ESRD), liver cirrhosis, human immunodeficiency virus infection (HIV) or acquired immunodeficiency syndrome (AIDS), laboratory data including LDL-C and C-reactive protein (CRP; particle enhanced immunoturbidimetric assay; Roche Diagnostics) concentrations, and outcomes, including the need for invasive mechanical ventilation, death, and dis-

charged alive. All laboratory tests refer to the first available results in the index admission or within six months prior to admission. The data were processed and analyzed without any personal identifiers to maintain patient confidentiality, as per the Health Insurance Portability and Accountability Act (HIPAA).

### 2.3. Exposure of Interest and Outcomes

The exposure of interest was LDL-C level. The primary endpoint was in-hospital death. The secondary endpoint was invasive mechanical ventilation.

### 2.4. Statistical Analysis

In order to create comparable groups, propensity score matching was performed, wherein propensity matching scores were calculated using a logistic regression model with thirteen covariates: age, sex, BMI, hypertension, diabetes, heart failure, coronary artery disease, peripheral arterial disease, atrial fibrillation, stroke, CKD or ESRD, COPD, and asthma. A nearest-neighbor matching without replacement (one-to-one) was conducted. [13] Once a patient with an available pre-COVID-19 LDL-C measurement (*n*= 398) had been matched with a patient who had an available LDL-C measurement during COVID-19 hospitalization (*n* = 2622), the latter was no longer available as a potential match for subsequent patients with available pre-COVID-19 LDL-C levels. A caliper width of 0.2 of the pooled standard deviation of the logit of the propensity score was used, which has been found to be optimal [14].

Continuous variables were presented as medians with interquartile range (IQR), and categorical variables as absolutes and relative frequencies. LDL-C levels were presented as means with standard deviations (SDs). The T-test was used to compare continuous variables, and the chi-square test was used for categorical variables. To further evaluate the balance of variables between groups of patients with available pre-COVID-19 LDL-C measurements and those with available LDL-C levels during COVID-19 hospitalization, both before and after propensity-score matching and standardized mean differences (SMDs) were computed [15]. A < 10%, in absolute terms, SMD would support the assumption of equally balanced groups.

After one-to-one matching in the derived cohort, analyses were conducted on a total of 796 patients (*n* = 398 and 398 patients); mortality outcomes and the need for invasive mechanical ventilation were assessed using logistic regression models, resulting in odds ratios (ORs) with corresponding 95% confidence intervals. We applied univariate analyses for each outcome, including different versions of LDL-C, and multivariate models for baseline characteristics found significant (*p* < 0.05) in the univariate analysis, for three different panels of data. Panel A included the total after the one-to-one matching cohort (*n* = 796), Panel B included the cohort of patients with available pre-COVID-19 LDL-C measurements (*n* = 398), and Panel C included patients who had an available LDL-C measurement during COVID-19 hospitalization after the one-to-one matching (*n* = 398). A *p*-value < 0.05 was considered statistically significant, and the statistical analysis was performed with STATA 14.

## 3. Results

### 3.1. Baseline Patient Characteristics

In total, 3020 patients were included in the analysis. Baseline characteristics of the overall and matched cohorts are presented in Table 1. As indicated by the different distributions of the propensity scores in both cohorts, before matching, baseline characteristics were different (Figure 1). Further, the standardized mean difference criterion revealed that before matching, 8 baseline covariates had a standardized mean difference < 0.10, while after matching there were 13. The median age was 61 years (IQR 50–72). Of the patients, 1920 (63.5%) were men and 1100 (36.5%) were women. The median BMI was 28.1 (IQR 24.5–32.5) kg/m$^2$. Diabetes, hypertension, and CKD were the most common comorbidities observed in 35.7%, 16.5%, and 7.2% of our patients, respectively. A total of 398 patients had available pre-COVID-19 LDL-C, with a median LDL-C of 75.8 mg/dL. A total of 2622 patients had available LDL-C during COVID-19 hospitalization, with a median LDL-C of 71 mg/dL (*p* = 0.011). The matched cohorts included 796 patients: 398 patients in each subgroup. Mean LDL-C was 81.10 mg/dL

and 75.37 mg/dL in the subgroups with available pre-COVID-19 LDL-C and available LDL-C during COVID-19 hospitalization, respectively (*p* = 0.044). There were no missing data.

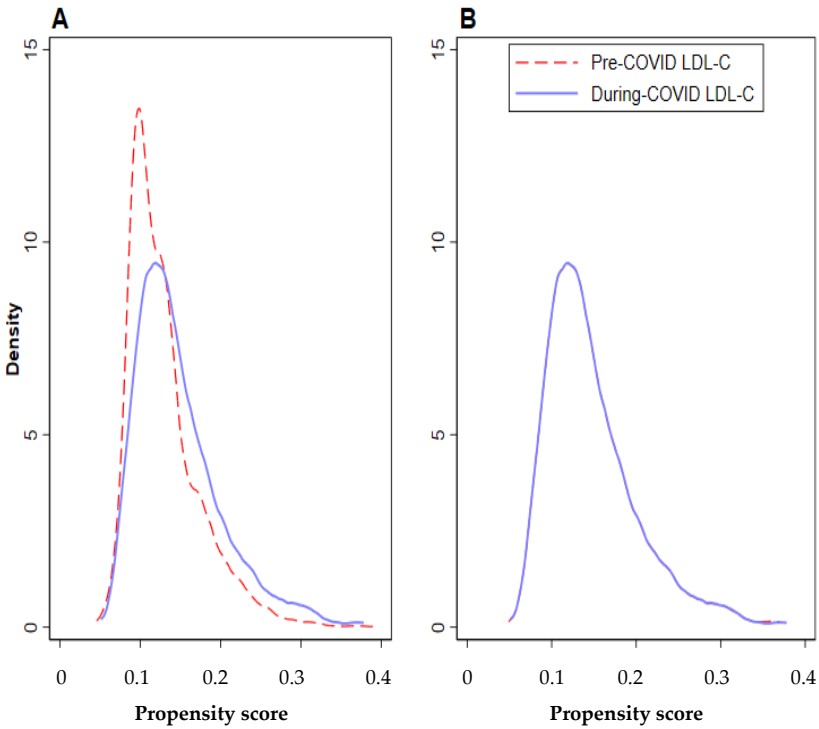

**Figure 1.** Propensity scores conditional on baseline covariate values (**A**) before matching and (**B**) after matching.

**Table 1.** Baseline characteristics.

| | Before Matching | | | | | After Matching | | | | |
|---|---|---|---|---|---|---|---|---|---|---|
| | **Total** | **Pre-COVID-19 LDL-C** | **During COVID-19 LDL-C** | | | **Total** | **Pre-COVID-19 LDL-C** | **During COVID-19 LDL-C** | | |
| **Variable** | *n* = 3020 | *n* = 398 | *n* = 2622 | *p*-Value | SMD | *n* = 796 | *n* = 398 | *n* = 398 | *p*-Value | SMD |
| Female n (%) | 1100 (36.4) | 166 (41.7) | 934 (35.6) | 0.019 | 0.125 | 330 (41.4) | 166 (41.7) | 164 (41.2) | 0.886 | 0.010 |
| Age (years) median (IQR) | 61 (50–72) | 62 (51–73) | 61 (50–71) | 0.087 | 0.092 | 63 (51.5–7) | 62.5 (51–73) | 63.5 (52–74) | 0.654 | 0.031 |
| BMI (median (IQR)) | 28.1 (24.5–32.5) | 27.4 (23.3–31.3) | 28.2 (24.7–32.7) | 0.003 | 0.159 | 27.7 (23.8–31.6) | 27.4 (23.3–31.3) | 27.9 (24.5–31.9) | 0.296 | 0.073 |
| Comorbidities n (%) | | | | | | | | | | |
| Hypertension | 497 (16.5) | 93 (23.3) | 404 (15.4) | <0.001 | 0.202 | 190 (23.8) | 93 (23.4) | 97 (24.4) | 0.739 | 0.023 |
| Diabetes | 1079 (35.7) | 183 (45.9) | 896 (34.2) | <0.001 | 0.242 | 368 (46.2) | 183 (45.9) | 185 (46.5) | 0.887 | 0.010 |
| Heart failure | 155 (5.1) | 27 (6.7) | 128 (4.9) | 0.109 | 0.081 | 53 (6.7) | 27 (6.8) | 26 (6.5) | 0.887 | 0.010 |
| CAD | 70 (2.3) | 13 (3.3) | 57 (2.2) | 0.177 | 0.067 | 30 (3.8) | 13 (3.3) | 17 (4.3) | 0.457 | 0.052 |
| PAD | 9 (0.3) | 2 (0.5) | 7 (0.3) | 0.422 | 0.038 | 5 (0.6) | 2 (0.5) | 3 (0.8) | 0.654 | 0.031 |
| Afib | 120 (3.9) | 21 (5.3) | 99 (3.8) | 0.153 | 0.072 | 45 (5.6) | 21 (5.3) | 24 (6) | 0.645 | 0.032 |
| Stroke | 106 (3.5) | 16 (4.0) | 90 (3.4) | 0.553 | 0.031 | 33 (4.2) | 16 (4) | 17 (4.3) | 0.859 | 0.012 |
| CKD/ESRD | 217 (7.2) | 47 (11.8) | 170 (6.5) | <0.001 | 0.185 | 97 (12.2) | 47 (11.8) | 50 (12.6) | 0.745 | 0.023 |
| COPD | 39 (1.3) | 7 (1.8) | 32 (1.2) | 0.375 | 0.044 | 12 (1.5) | 7 (1.8) | 5 (1.3) | 0.561 | 0.041 |
| Asthma | 81 (2.7) | 15 (3.8) | 66 (2.5) | 0.150 | 0.071 | 32 (4) | 15 (3.8) | 17 (4.3) | 0.718 | 0.025 |
| HIV or AIDS | 89 (2.9) | 23 (5.8) | 66 (2.5) | <0.001 | 0.163 | 33 (4.2) | 23 (5.8) | 10 (2.5) | 0.021 | 0.164 |
| Cirrhosis | 22 (0.7) | 6 (1.5) | 16 (0.6) | 0.050 | 0.087 | 8 (1.) | 6 (1.5) | 2 (0.5) | 0.155 | 0.100 |
| LDL-C | | | | | | | | | | |
| Mean (SD) | 76.5 (0.70) | 81.1 (1.92) | 75.84 (0.8) | 0.011 | 0.136 | 78.24 (1.42) | 81.10 (1.92) | 75.37 (2.10) | 0.044 | 0.142 |
| Median (IQR) | 71.6 (49–98.5) | 75.8 (54–103) | 71.00 (48–98) | | | 72.70 (49.9–101) | 75.80 (54–103) | 68.3 (46–97) | | |
| LDL-C n (%) | | | | 0.019 | 0.126 | | | | 0.011 | 0.181 |
| ≤70 | 1463 (48.4) | 171 (42.9) | 1292 (49.3) | | | 378 (47.5) | 171 (42.9) | 207 (52) | | |
| >70 | 1557 (51.5) | 227 (57) | 1330 (50.7) | | | 418 (52.5) | 227 (57) | 191 (47.9) | | |

Abbreviations and symbols: SMD = standard median deviation; COVID-19 = Coronavirus; BMI = body mass index in kg/m²; kg = kilogram; m = meter; N = number; IQR = interquartile range; LDL-C = low density lipoprotein cholesterol (in mg/dL); mg = milligram; dL = deciliter; CAD = coronary artery disease; PAD = peripheral artery disease; Afib = atrial fibrillation; CKD: chronic kidney disease; ESRD = end-stage renal disease; COPD = chronic obstructive pulmonary disease; HIV = human immunodeficiency virus; AIDS = acquired immunodeficiency disease syndrome.

### 3.2. Outcomes

In the overall cohort of 3020 patients, 643 (21.3%) died and 313 (10.4%) were intubated and required invasive mechanical ventilation. Mean LDL-C was 61.79 mg/dL in patients that died, 63.41 mg/dL in the patients that required intubation, and 80.88 mg/dL in the patients that survived without requiring intubation. Outcomes and LDL-C levels per outcome are presented in Table 2.

**Table 2.** Outcomes and LDL-C levels per outcome.

| | Before Matching | | | | | After Matching | | | | |
|---|---|---|---|---|---|---|---|---|---|---|
| | Total | Pre-COVID-19 Available LDL-C | During COVID-19 Available LDL-C | | | Total | Pre-COVID-19 Available LDL-C | During COVID-19 Available LDL-C | | |
| Outcomes | *n* = 3020 | *n* = 398 | *n* = 2622 | *p*-Value | SMD | *n* = 796 | *n* = 398 | *n* = 398 | *p*-Value | SMD |
| Death no. (%) | | | | 0.251 | 0.062 | | | | 1.000 | 0.000 |
| No | 2377 (78.7) | 322 (80.9) | 2055 (78.4) | | | 644 (80.9) | 322 (80.9) | 322 (80.9) | | |
| Yes | 643 (21.2) | 76 (19.1) | 567 (21.6) | | | 152 (19.1) | 76 (19.1) | 76 (19.1) | | |
| IMV no. (%) | | | | 0.001 | 0.202 | | | | 0.042 | 0.144 |
| No | 2707 (89.6) | 376 (94.5) | 2331 (88.9) | | | 737 (92.6) | 376 (94.5) | 361 (90.7) | | |
| Yes | 313 (10.4) | 22 (5.5) | 291 (11.1) | | | 59 (7.4) | 22 (5.5) | 37 (9.3) | | |
| **LDL-C** | | | | | | | | | | |
| **Death** | | | | | | | | | | |
| No | *n* = 2377 | *n* = 322 | *n* = 2055 | | | *n* = 644 | *n* = 322 | *n* = 322 | | |
| Mean (SD) | 80.5 (0.8) | 81.11 (2.2) | 80.43 (0.8) | 0.763 | 0.017 | 80.41 (1.6) | 81.1 (2.2) | 79.7 (2.3) | 0.654 | 0.035 |
| Median (IQR) | 75.6 (53–102.4) | 75 (53.8–103) | 75.8 (53–102.2) | | | 74 (52.1–102.2) | 75 (53.8–103) | 73.6 (51–101.4) | | |
| LDL-C no. (%) | | | | 0.655 | 0.026 | | | | 0.475 | 0.056 |
| ≤70 | 1046 (44) | 138 (42.9) | 908 (44.2) | | | 285 (44.3) | 138 (42.9) | 147 (45.7) | | |
| >70 | 1331 (55.9) | 184 (57.1) | 1147 (55.8) | | | 359 (57.1) | 184 (57.1) | 175 (54.3) | | |
| Yes | *n* = 643 | *n* = 76 | *n* = 567 | | | *n* = 152 | *n* = 76 | *n* = 76 | | |
| Mean (SD) | 61.8 (1.4) | 81 (4.24) | 59.21 (1.5) | <0.001 | 0.606 | 69 (3.3) | 81 (4.2) | 57 (4.62) | <0.001 | 0.621 |
| Median (IQR) | 57 (37–82) | 77 (56–103.3) | 55 (35.8–8) | | | 62.1 (38.7–87.2) | 77 (55.9–103.3) | 47 (32.8–66.9) | | |
| LDL-C no. (%) | | | | <0.001 | 0.502 | | | | <0.001 | 0.777 |
| ≤70 | 417 (64.9) | 33 (43.4) | 384 (67.7) | | | 93 (61.2) | 33 (43.4) | 60 (78.9) | | |
| >70 | 226 (35.2) | 43 (56.6) | 183 (32.3) | | | 59 (38.8) | 43 (56.6) | 16 (21) | | |
| **IMV** | | | | | | | | | | |
| No | *n* = 2707 | *n* = 376 | *n* = 2331 | | | *n* = 737 | *n* = 376 | *n* = 361 | | |
| Mean (SD) | 78.05 (0.7) | 81.44 (1.9) | 77.5 (0.8) | 0.061 | 0.103 | 79.23 (1.5) | 81.44 (1.9) | 79.2 (1.5) | 0.124 | 0.113 |
| Median (IQR) | 73.6 (51–99.8) | 75 (54–103.4) | 73 (50.6–9) | | | 73 (51–102) | 75 (54–103.4) | 71 (48–99) | | |
| LDL-C no. (%) | | | | 0.214 | 0.069 | | | | 0.162 | 0.103 |
| ≤ 70 | 1261 (46.6) | 164 (43.6) | 1097 (47) | | | 340 (46.1) | 164 (43.6) | 176 (48.8) | | |
| > 70 | 1446 (53.4) | 212 (56.4) | 1234 (52.9) | | | 397 (53.9) | 212 (56.4) | 185 (51.3) | | |
| Yes | *n* = 313 | *n* = 22 | *n* = 291 | | | *n* = 59 | *n* = 22 | *n* = 37 | | |
| Mean (SD) | 63.41 (2.3) | 75.29 (7.5) | 62.5 (2.4) | 0.156 | 0.334 | 65.78 (5.8) | 75.3 (7.5) | 60.13 (7.9) | 0.207 | 0.357 |
| Median (IQR) | 55.2 (83–37) | 80.1 (47–91.2) | 54.4 (36–82.6) | | | 57 (37–8) | 80.1 (47–91.2) | 47 (34.2–6) | | |
| LDL-C no. (%) | | | | 0.001 | 0.742 | | | | <0.001 | 1.213 |
| ≤70 | 202 (64.5) | 7 (31.8) | 195 (67) | | | 38 (64.4) | 7 (31.8) | 31 (83.8) | | |
| >70 | 111 (35.5) | 15 (68.2) | 96 (32.9) | | | 21 (35.6) | 15 (68.2) | 6 (16.2) | | |

Abbreviations and symbols: N = number; LDL-C = low density lipoprotein cholesterol (in mg/dL); mg = milligram; dL = deciliter; IQR = interquartile range, SD = standard deviation; IMV = invasive mechanical ventilation.

In the matched cohort of 796 patients, 152 (19.1%) died (pre-COVID-19 LDL-C available cohort: 76 (19.1%); during COVID-19 LDL-C cohort: 76 (19.1%); *p* = 1.000) and 59 (7.41%) required intubation (pre-COVID-19 LDL-C available cohort: 22 (5.53%); during COVID-19 LDL-C cohort: 37 (9.30%), *p* = 0.042).

### 3.3. Logistic Regression Analyses

Mortality

In the matched cohort (796 patients), from multivariate logistic regression analysis, LDL was found to be inversely associated with in-hospital death (OR 0.99, 95% CI 0.986–0.999, *p* = 0.036) (Table 3, panel A). Specifically, when LDL-C was analyzed as a categorical variable,

LDL > 70 mg/dL was associated with a 48% lower likelihood of in-hospital death (OR 0.52, 95% CI 0.35–0.78, $p < 0.001$) (Table 3, panel A). A similar association was noted in the subgroup of patients with available LDL during COVID-19 admission (398 patients) from the matched cohort (OR 0.98, 95% CI 0.97–0.99, $p = 0.006$) (Table 3, panel C). In contrast, the same analysis in the subgroup of patients with available pre-COVID-19 LDL levels (398 patients) from the matched cohort revealed no significant association (OR 1.00, 95% CI 0.99–1.01, $p = 0.833$) (Table 3, panel B).

**Table 3.** Multivariate logistic regression analyses for death in the cohort after matching.

| | Model 1 | Model 2 |
|---|---|---|
| **PANEL A: Total Cohort ($n = 796$)** | **OR, 95% CI, $p$-Value** | **OR, 95% CI, $p$-Value** |
| Female | 0.61 * (0.40–0.94) $p = 0.026$ | 0.63 * (0.41–0.98) $p = 0.040$ |
| Age per 10 years | 1.44 ** (1.25–1.67) $p < 0.001$ | 1.45 ** (1.25–1.67) $p < 0.001$ |
| BMI | 1.07 ** (1.04–1.11) $p < 0.001$ | 1.07 ** (1.04–1.11) $p < 0.001$ |
| LDL-C | 0.99 * (0.986–0.999) $p = 0.036$ | |
| LDL-C > 70 | | 0.52 ** (0.35–0.78) $p = 0.001$ |
| **PANEL B: Pre-COVID-19 LDL-C cohort ($n = 398$)** | | |
| Female | 0.64 (0.35–1.18) $p = 0.154$ | 0.64 (0.35–1.18) $p = 0.156$ |
| Age per 10 years | 1.65 ** (1.32–2.05) $p < 0.001$ | 1.64 ** (1.32–2.04) $p < 0.001$ |
| BMI | 1.06 * (1.01–1.10) $p = 0.010$ | 1.06 * (1.01–1.10) $p = 0.011$ |
| LDL-C | 1.00 (0.99–1.01) $p = 0.833$ | |
| LDL-C > 70 | | 0.96 (0.54–1.71) $p = 0.897$ |
| **PANEL C: During COVID-19-LDL-C cohort ($n = 398$)** | | |
| Female | 0.52 * (0.28–0.98) $p = 0.042$ | 0.57 (0.31–1.08) $p = 0.084$ |
| Age per 10 years | 1.26 * (1.04–1.53) $p = 0.017$ | 1.27 * (1.04–1.55) $p = 0.018$ |
| BMI | 1.10 ** (1.05–1.16) $p < 0.001$ | 1.10 ** (1.04–1.15) $p < 0.001$ |
| LDL-C | 0.98 ** (0.97–0.99) $p = 0.006$ | |
| LDL-C > 70 | | 0.22 ** (0.12–0.42) $p < 0.001$ |

Notes: LDL-C was run as a continues variable in model 1 and as a dichotomous variable in model 2; models adjusted for significant variables at 5% in univariate analysis; panel A: adjusted for HTN, DM, CKD or ESRD, and asthma; panel B: adjusted for HTN and DM; panel C: adjusted for HTN, DM, CKD or ESRD; ** $p < 0.01$, * $p < 0.05$; The nonsignificant ORs for the comorbidities have not been listed. BMI in kg/m$^2$; LDL-C in mg/dL. Abbreviations and symbols: BMI = body mass index; kg = kilogram; m = meter; LDL-C = low density lipoprotein; mg = milligram; dL = deciliter; HTN = hypertension; DM = diabetes mellitus; CKD: chronic kidney disease; ESRD = end-stage renal disease.

### 3.4. Invasive Mechanical Ventilation

In the matched cohort (796 patients), on multivariate logistic regression analysis, no significant association was found between LDL level and invasive mechanical ventilation, when taken as a continuous variable (OR 0.99, 95% CI 0.98–1.00, $p = 0.065$) (Table 4, panel A). When LDL was analyzed as a categorical variable, LDL > 70 mg/dL was associated with 47% lower likelihood of invasive mechanical ventilation (OR 0.53, 95% CI 0.29–0.95, $p = 0.034$) (Table 4, panel A). In the subgroup of patients with available LDL levels during COVID-19 admission (398 patients), from the matched cohort, no variation was observed when analyzed as a continuous variable. However, a significant difference was observed when LDL was analyzed as a categorical variable (OR 0.18, 95% CI 0.07–0.47, $p < 0.001$) (Table 4, panel C). The same analysis in the subgroup of patients with available pre-COVID-19 LDL (398 patients) of the matched cohort revealed no significant association, whether LDL was analyzed as a continuous or categorical variable (OR 1.00, 95% CI 0.99–1.01, $p = 0.833$) (Table 4, panel B).

**Table 4.** Multivariate logistic regression analyses for invasive mechanical ventilation in the cohort after matching.

| | Model 1 | Model 2 |
|---|---|---|
| **PANEL A: Total Cohort ($n = 796$)** | **OR, 95% CI, $p$-Value** | **OR, 95% CI, $p$-Value** |
| Female | 0.74 (0.41–1.34) $p = 0.317$ | 0.75 (0.41–1.36) $p = 0.342$ |
| Age per 10 years | 1.13 (0.92–1.40) $p = 0.248$ | 1.15 (0.92–1.42) $p = 0.215$ |
| BMI | 1.04 * (1.00–1.08) $p = 0.039$ | 1.04 * (1.00–1.07) $p = 0.049$ |
| LDL-C | 0.99 (0.98–1.00) $p = 0.065$ | |
| LDL-C > 70 | | 0.53 * (0.29–0.95) $p = 0.034$ |
| **PANEL B: Pre-COVID-19 LDL-C cohort ($n = 398$)** | | |
| Female | 1.46 (0.60–3.54) $p = 0.404$ | 1.45 (0.60–3.52) $p = 0.414$ |
| Age per 10 years | 1.40 * (1.03–1.92) $p = 0.033$ | 1.44 * (1.05–1.96) $p = 0.024$ |
| BMI | 1.01 (0.96–1.06) $p = 0.697$ | 1.01 (0.96–1.06) $p = 0.679$ |
| LDL-C | 1.00 (0.99–1.01) $p = 0.543$ | |
| LDL-C > 70 | | 1.81 (0.71–4.57) $p = 0.212$ |

**Table 4.** *Cont.*

|  | Model 1 | Model 2 |
|---|---|---|
| **PANEL A: Total Cohort (*n* = 796)** | **OR, 95% CI, *p*-Value** | **OR, 95% CI, *p*-Value** |
| **PANEL C: During COVID-19-LDL-C cohort (*n* = 398)** |  |  |
| Female | 0.39 * (0.17–0.91) *p* = 0.029 | 0.45 (0.19–1.03) *p* = 0.059 |
| Age per 10 years | 0.91 (0.71–1.16) *p* = 0.443 | 0.89 (0.70–1.14) *p* = 0.352 |
| BMI | 1.06 * (1.01–1.11) *p* = 0.015 | 1.05 * (1.01–1.10) *p* = 0.018 |
| LDL-C | 0.99 (0.97–1.00) *p* = 0.101 |  |
| LDL-C > 70 |  | 0.18 ** (0.07–0.47) *p* < 0.001 |

Notes: LDL-C was run as a continuous variable in model 1 and as a dichotomous variable in model 2; models adjusted for significant variables at 5% for the univariate analysis; panel A was adjusted for HTN; no significant univariate associations in panels B and C; ** *p* < 0.01, * *p* < 0.05; the non-significant ORs for the comorbidities have not been listed. BMI in kg/m$^2$; LDL-C in mg/dL. Abbreviations and symbols: BMI = body mass index; kg = kilogram; m= meter; LDL-C = low density lipoprotein; mg = milligram; dL = deciliter; HTN = hypertension.

## 4. Discussion

Our study investigated the association between the levels of pre-COVID-19 LDL-C and levels of LDL-C during COVID-19 hospitalization with the need for invasive mechanical ventilation and death in patients hospitalized with COVID-19 in NYC Health and Hospitals, the largest public health system in the country. The key findings of our study can be summarized as follows: (1) lower LDL-C level measured during COVID-19 hospitalization was associated with a higher likelihood of invasive mechanical ventilation and in-hospital death, and (2) no association was noted between pre-COVID-19 LDL-C level and invasive mechanical ventilation or death.

The findings of our study are significant and unique in comparison to the currently available literature. We demonstrated an inverse relationship between LDL-C levels obtained during COVID-19 hospitalization with the likelihood of invasive mechanical ventilation and death in patients with COVID-19. Additionally, we found a progressive decrease in LDL-C levels with increasing severity of COVID-19. This is a paradoxical finding, as a high comorbidity burden is a well-established risk factor for poor outcomes of COVID-19 [1–8]. The findings of our study are comparable to those of previously conducted studies, wherein low LDL-C levels have been associated with higher mortality and worse outcomes in patients with COVID-19 [9,10,16–18]. Conducted early in the pandemic, Wei et al., in their analysis of 597 patients with COVID-19, observed an inverse relationship between LDL-C levels and COVID-19 severity [17]. Similarly, Sun et al. found lower LDL-C concentrations among patients with severe COVID-19, including nonsurvivors, compared to patients with milder COVID-19 [19]. Although there are variations in the extent of the drop in LDL-C levels, it is appreciable that LDL-C levels tend to fall during the acute phase of COVID-19.

Lipids are an integral part of the pathophysiology and disease course of COVID-19, as those with cardiovascular comorbidities, particularly those with dyslipidemia, are at a higher risk of developing severe COVID-19 [20–25]. Lower LDL-C levels observed during COVID-19 can be attributed to three main pathophysiologic mechanisms: (i) decreased LDL-C production and lipid absorption, (ii) decreased cholesterol transport, (iii) and the inflammatory reaction pathognomonic of COVID-19 [17]. Studies have shown that the SARS-CoV-2 virus may lead to decreased LDL-C levels by causing direct damage to the liver, and hampering lipid biosynthesis. This is usually characterized by a moderate increase in ALT, AST, and ALP [17,26,27]. However, in the absence of transaminitis, low LDL-C levels may be secondary to decreased gut absorption of lipids [26]. Moreover, COVID-19 leads to an increase in the proinflammatory cytokines IL-6, TNF α, and IL-1 β. These cytokines are known to modulate lipid metabolism by decreasing cholesterol efflux and transport [27,28]. Other possible mechanisms include tissue lipoprotein lipase inhibition, increased synthesis of hepatic triglyceride, loss of synthesis–utilization balance, free radical-mediated lipid degeneration, and leakage of cholesterol molecules into alveolar spaces due to increased vascular permeability [17,29]. Apart from sepsis, a similar finding of decreased LDL-C levels has been reported in other viral illnesses, such as HIV infection (human immunodeficiency virus), Dengue, and EBV infection (Epstein–Barr virus) [27,30,31]. Cholesterol is a major component of the viral cell membrane, and viruses are completely dependent

on the host cell machinery for replication and metabolism. Exploiting this knowledge, lipid pathway-blocking drugs have been explored as a treatment option for various viral infections [32]. Makris et al. deduced, via a randomized control trial, that the use of pravastatin was associated with decreased frequency of ventilator-associated pneumonia and increased survival among the treated group in comparison to the untreated group [33]. Adding on to this information, Tleyjeh et al. and Douglas et al. also reported that statin was associated with reduced infection and six-month all-cause mortality after pneumonia, respectively [34,35]. Statins have been explored to be a potential drug for the treatment and prevention of COVID-19. Zhang X.J. et al. and Rodriguez- Nava G. et al. reported a favorable recovery profile and reduced risk of mortality in patients with COVID-19 [36,37]. Daniels L.B. et al. reported a reduced risk of developing severe COVID-19 infection and a faster recovery time in individuals taking statin 30 days before COVID-19 hospitalization [38]. It is also important to note that lipid pathway-blocking drugs can interact with the antiviral agents used in the treatment of COVID-19, such as remdesivir. Both of these drugs are substrates for cytochrome P450 3A4 (CYP3A4), which might lead to interactions and adverse effects upon concurrent administration [39]. Since we lacked data on lipid-lowering drug usage by the patients during the pre-COVID-19 and COVID-19 periods, we are unable to comment on the impact these drugs had on the outcomes of our study.

Alterations in LDL-C levels in the acute phase of COVID-19 infection make it unreliable for assessing the baseline lipid profile and cardiovascular risk stratification during a period of active infection. This is evident by the fact that we found no association between pre-COVID-19 LDL-C levels and death, despite matching both cohorts for age, sex, BMI, and twelve baseline comorbidities. Hence, we can safely conclude that lipid profiles should not be obtained during the acute phase of COVID-19 to diagnose dyslipidemia or for cardiovascular risk stratification. Additionally, considering the availability of other reliable and well-studied markers of inflammation, such as C-reactive protein (CRP), D-dimer, and Interleukin-6, to assess the severity of COVID-19 [40–42], the practical utility of LDL-C for routine prognostication of COVID-19 severity is unclear.

*Strengths and Limitations*

Our study has several strengths. First, our patient population consists of ethnically diverse and socioeconomically disadvantaged individuals who are often underrepresented in the published literature. Second, we conducted a robust statistical analysis using the propensity-matched scoring system. However, we acknowledge that our study is not devoid of notable limitations. Being a retrospective study conducted by review of electronic medical records, there is a possibility for observational bias and unmeasured confounding factors not entered in the EMR system, such as ethnicity, nationality, nutritional status, data on the days after the onset of illness, etc., which cannot be mitigated by a propensity-matched scoring system. Despite a thorough search, one or more relevant electronic medical records may have been missed. However, we employed an exhaustive and independent review process, and followed a strict methodology to minimize these biases. Second, since we did not obtain follow-up lipid profiles on survivors, we were unable to determine the time taken for the lipid panel to return to the baseline post-acute phase of COVID-19. Third, our study lacked data on superimposed bacterial infections, such as *Pseudomonas aeruginosa* or other concomitant viral infections [43]. Data on the use of other lipid pathway-blocking drugs were also not collected. These unmeasured variables could have influenced the outcomes of our study.

## 5. Conclusions

Low LDL-C levels measured during COVID-19 hospitalization were associated with a higher likelihood of invasive mechanical ventilation and in-hospital death. A similar association was not found when comparing pre-COVID-19 LDL-C levels with the likelihood of invasive mechanical ventilation and death from COVID-19. The results of our study suggest that LDL-C levels obtained during COVID-19 might not be an accurate and reliable indicator of the baseline lipid profile and cardiovascular disease risk.

**Author Contributions:** All authors contributed substantially to the preparation of the manuscript. Conceptualization, L.P., D.K. and A.M.; methodology, D.K., L.P., A.M., A.K., S.X. and S.N.; software, S.N., L.P., Y.E.D. and C.J.B.M.; validation, D.V., Y.E.D., A.M., A.K., D.B., M.A.D.A. and S.V.; formal analysis, D.K.; investigation, S.J.A. and M.A.D.A.; resources, L.P. and D.K.; data curation, S.N., Y.E.D., C.J.B.M., S.V., P.Z. and S.J.A.; writing—original draft preparation, A.M., L.P., S.N., A.K., S.J.A., D.B., M.A.D.A., Y.E.D., C.J.B.M., S.V., P.Z. and D.V.; writing—review and editing, A.K., S.N., S.J.A., D.B., M.A.D.A., Y.E.D., L.P., D.K., S.X., D.V. and C.J.B.M.; visualization, D.B., M.A.D.A., D.V., C.J.B.M., S.V., S.N., P.Z., S.X., D.K. and S.J.A.; supervision, L.P., D.V. and S.X.; project administration, L.P., S.N. and A.K. All authors have read and agreed to the published version of the manuscript.

**Funding:** This research did not receive any specific grant from funding agencies in the public, commercial, or not-for-profit sectors.

**Institutional Review Board Statement:** Biomedical Research Alliance of New York (BRANY) Institutional Review Board with a waiver of informed consent (IRB# 20-12-103-373).

**Informed Consent Statement:** Data were fully deidentified and anonymized before they were accessed, hence the IRB waived the requirement for informed consent.

**Data Availability Statement:** The data presented in this study are available on request from the corresponding author. The data are not publicly available to protect patient confidentiality.

**Conflicts of Interest:** The authors declare no conflict of interest.

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
