# Peer review of "Invasive Mechanical Ventilation and Death Was More Likely in Patients with Lower LDL Cholesterol Levels during COVID-19 Hospitalization: A Retrospective Propensity-Matched Cohort Study"

_2673-527X, doi:10.3390/jor3020005_

Round 1
Reviewer 1 Report
The authors conducted a retrospective observational study of patients with COVID-19 comparing pre-COVID-19 LDL-C levels or LDL-C levels obtained during COVID-19 with the need for invasive mechanical ventilation and death. It found that low LDL cholesterol measured during COVID-19 was associated with a higher likelihood of invasive mechanical ventilation and in-hospital death. No such association was found between pre-COVID-19 LDL-C and these outcomes.
1. Why was 70 mg/d chosen as the cut-off value if mean LDL-C was 81.10 mg/dL and 75.37 mg/dL in the subgroups with available pre-COVID-19 LDL-C and available LDL-C during COVID-19 hospitalization, respectively?
2. Why are comorbidities not taken into account in the regression analysis? The authors only included gender, age, BMI, and LDL-C.
3. Nationality was not taken into account when building the model?
4. On what day after the onset of the disease was hospitalization carried out and at what moment was LDL-C determined? Are the values comparable across the cohort of patients, and does later hospitalization affect outcome, for example?
Reviewer 2 Report
Thank you for the opportunity to review this manuscript. There are some strengths but also several issues with the present propensity-matched cohort study. Please see below for my comments.
Specific comments:
1. The study title is confusing and says nothing about the study design. Suggest mentioning in the title that this was a retrospective propensity-matched cohort study.
2. "Hyperlipidemia, like other cardiovascular risk factors, has been associated with worse outcomes in patients with Coronavirus Disease 2019 (COVID-19) [1-6]" - some of the references here are more than 2 years old, suggesting updating the studies as our understanding of COVID-19 has changed rather significantly compared to early in the pandemic. Several studies using the UK Biobank and other databases were not cited.
3. "Importantly, majority of these studies were conducted in populations outside the United States" - I am not sure this is really an important point to make as European countries and the UK also have an ethnically diverse and socioeconomically disadvantaged patient population. I suggest authors speak more directly about the rationale of the study and its intended contribution to the broader literature.
4. "... March 1, 2020 to October 31, 2020" - is there a reason why this particular study period was chosen? This period was rather short and significantly dated as it is more than two years ago. How should the period of study be considered? Does the choice of time period affect the results of the study? If so, what is the impact?
5. Please change "... accessed the IRB waived" to "... accessed, hence the IRB waived".
6. "... derived cohort (N=398 & 398 patients)" - is there a missing word here?
7. More information should be provided on the propensity matching process, e.g. whether authors identified imbalances in strong risk factors and in common covariates etc.
8. Some comments on the local COVID-19 situation and hospital capacity on the ground during the period the data was conducted would be helpful as well.
9. Patients with a variety of different infections (gram positive bacterial, gram negative bacterial, viral, tuberculosis, parasites) have similar alterations in plasma lipid levels. Specifically, total cholesterol, LDL-C, and HDL-C levels are decreased while plasma triglyceride levels may be elevated or inappropriately normal for the poor nutritional status. Ultimately, it should be noted that these observations are subject to the caveats of confounding variables and reverse causation effecting the results.
10. Although not available during the study period, the authors should also mention that certain antivirals that are used to treat COVID-19 infections may interact with lipid lowering drugs. Remdesivir and Paxlovid (nirmatrelvir and ritonavir) are metabolized by the CYP3A4 pathway and statins e.g. atorvastatin, simvastatin, and lovastatin, are also metabolized by this pathway.
Reviewer 3 Report
In their work, the authors attempted to assess the relationship between LDL concentration and worse outcomes of patients with COVID-19.
The result, in a way surprising, is exciting and informs us that the measurement of LDL during severe forms of the disease may not be the best indicator of lipid metabolism.
I am asking authors to consider the following issues:
1. whether the result of the analysis may be affected by the use of statins by patients before the onset of COVID-19,
2. please extend the references with the latest publications on the subject addressed by the authors:
doi: 10.3390/vaccines11010108.
doi: 10.3390/ijms21103544.
Round 2
Reviewer 1 Report
I have no more comments on the article, the authors have responded to those made earlier and revised the manuscript. I think that in its present form the article can be recommended for publication.
Author Response
We thank the reviewer for the kind words and the opportunity to help improve our manuscript.
Reviewer 2 Report
Thank you for the revisions.
Specific comments:
1. "Third, our study lacked data on superimposed bacterial or other concomitant viral infection ..." - specifically, it should be mentioned that during the COVID-19 pandemic, several international reports found a slight increase in the incidence of P. aeruginosa bacteremia (citation: pubmed.ncbi.nlm.nih.gov/36830319). Furthermore, P. aeruginosa is an opportunistic pathogen with a predilection for immunocompromised patients, and for critically ill patients who require mechanical ventilation, P. aeruginosa is also the most common multidrug-resistant Gram-negative pathogen.
2. "The results of our study suggest that LDL-C levels obtained during COVID-19 might not be an accurate and reliable indicator of baseline lipid profile and cardiovascular disease risk due to alterations in the acute phase of infection" - do you mean LDL-C levels obtained during an acute COVID-19 infection? This is a rather confusing and misleading sentence.
